# Perspective on the Use of DNA Repair Inhibitors as a Tool for Imaging and Radionuclide Therapy of Glioblastoma

**DOI:** 10.3390/cancers14071821

**Published:** 2022-04-03

**Authors:** Liesbeth Everix, Shankari Nair, Cathryn H. S. Driver, Ingeborg Goethals, Mike M. Sathekge, Thomas Ebenhan, Charlot Vandevoorde, Julie Bolcaen

**Affiliations:** 1Molecular Imaging Center Antwerp (MICA), University of Antwerp, 2610 Wilrijk, Belgium; liesbeth.everix@uantwerpen.be; 2Radiation Biophysics Division, SSC Laboratory, iThemba LABS, Cape Town 7131, South Africa; snair@tlabs.ac.za (S.N.); cvandevoorde@tlabs.ac.za (C.V.); 3Pre-Clinical Imaging Facility, Nuclear Medicine Research Infrastructur e NOC, Pelindaba, Brits 0242, South Africa; cathryn.driver@necsa.co.za (C.H.S.D.); mike.sathekge@up.ac.za (M.M.S.); 4Department of Radiochemistry, South African Nuclear Energy Corporation, Pelindaba, Brits 0240, South Africa; 5Department of Nuclear Medicine, Ghent University Hospital, 9000 Ghent, Belgium; ingeborg.goethals@ugent.be; 6Nuclear Medicine Department, University of Pretoria and Steve Biko Academic Hospital, Pretoria 0001, South Africa

**Keywords:** targeted radionuclide therapy, glioblastoma, radiochemistry, theranostics, molecular imaging, DNA repair inhibitors, DNA damage, radiopharmaceuticals, nuclear medicine

## Abstract

**Simple Summary:**

The current routine treatment for glioblastoma (GB), the most lethal high-grade brain tumor in adults, aims to induce DNA damage in the tumor. However, the tumor cells might be able to repair that damage, which leads to therapy resistance. Fortunately, DNA repair defects are common in GB cells, and their survival is often based on a sole backup repair pathway. Hence, targeted drugs inhibiting essential proteins of the DNA damage response have gained momentum and are being introduced in the clinic. This review gives a perspective on the use of radiopharmaceuticals targeting DDR kinases for imaging in order to determine the DNA repair phenotype of GB, as well as for effective radionuclide therapy. Finally, four new promising radiopharmaceuticals are suggested with the potential to lead to a more personalized GB therapy.

**Abstract:**

Despite numerous innovative treatment strategies, the treatment of glioblastoma (GB) remains challenging. With the current state-of-the-art therapy, most GB patients succumb after about a year. In the evolution of personalized medicine, targeted radionuclide therapy (TRT) is gaining momentum, for example, to stratify patients based on specific biomarkers. One of these biomarkers is deficiencies in DNA damage repair (DDR), which give rise to genomic instability and cancer initiation. However, these deficiencies also provide targets to specifically kill cancer cells following the synthetic lethality principle. This led to the increased interest in targeted drugs that inhibit essential DDR kinases (DDRi), of which multiple are undergoing clinical validation. In this review, the current status of DDRi for the treatment of GB is given for selected targets: ATM/ATR, CHK1/2, DNA-PK, and PARP. Furthermore, this review provides a perspective on the use of radiopharmaceuticals targeting these DDR kinases to (1) evaluate the DNA repair phenotype of GB before treatment decisions are made and (2) induce DNA damage via TRT. Finally, by applying in-house selection criteria and analyzing the structural characteristics of the DDRi, four drugs with the potential to become new therapeutic GB radiopharmaceuticals are suggested.

## 1. Introduction

Treatment challenges posed by malignant gliomas remain considerable, and many derive from the molecular and cellular heterogeneity inherent to these tumor variants [1,2]. New treatment strategies for glioblastomas (GB), known as the most malignant gliomas (grade IV), are urgently warranted. For newly diagnosed GB patients with overall good health status, the standard of care includes maximal surgical resection, combined external beam radiation therapy (RT), and temozolomide (TMZ), followed by maintenance TMZ [3]. However, even with an optimal treatment protocol and recent advances in targeted therapies, survival has only slightly improved, and almost all tumors recur [1,4]. Molecular biomarkers play an increasing role in treatment decisions and response prediction. For example, the methylation status of the O^6^-methylguanine-DNA methyltransferase (MGMT) promoter is a major cause of TMZ resistance [5,6]. The focus for advancing GB therapy lies in the field of personalized, targeted therapy with the ultimate aim to selectively eradicate GB cells without damaging the surrounding healthy brain tissue [4,7,8]. To achieve this, mechanisms that induce therapy resistance and strategies to induce selective cell death in GB cells need to be explored and exploited. GB is recognized as being highly radioresistant and influenced by the presence of glioma stem cells (GSCs), cellular hypoxia, high cell heterogeneity, and aberrant activation of DNA damage response (DDR) proteins [9,10]. The dysregulation of the DDR in GB allows cancer cells to repair DNA damage and results in resistance to the current state-of-the-art therapies. In contrast to normal cells, components of the DDR pathway are frequently compromised in tumor cells, and their survival is often based on a sole backup pathway. Hence, targeted strategies against essential components of the DDR offer the possibility to promote cell death in cancer cells and increase the tumor’s sensitivity to cancer therapies based on the principle of ‘synthetic lethality’ (Figure 1) [10,11,12,13]. In order to sensitize GB cells to DNA damaging agents, two approaches can be adopted. First, directly targeting key DNA damage signaling kinases such as phosphatidylinositol 3′ kinase (PI3K)-related kinases (PIKKs) and PIKK-regulated downstream kinases. These include DNA damage sensor and repair proteins, e.g., ataxia-telangiectasia mutated (ATM), ATM-RAD3-related (ATR) protein, and DNA-dependent protein kinase (DNA-PK) [14]. Second, they interfere with cell cycle checkpoint proteins, which monitor DNA integrity before cell division (G2–M checkpoint) and DNA replication (G1–S checkpoint) [15]. Interestingly, ‘replication stress’ present in cancer cells could further be enhanced following these therapies through further loosening the remaining checkpoints and inducing failure of further proliferation [16].

In this review, the rationale and current status of targeted drugs that inhibit essential DDR kinases (DDRi) for the therapy of GB are given. Secondly, a perspective is given on radiopharmaceuticals for nuclear imaging and targeted radionuclide therapy (TRT) targeting DDR kinases. The ability to monitor DNA repair processes using nuclear imaging may be an asset for personalized GB therapy and for monitoring the response to DNA damaging treatments and DDRi. Finally, selection criteria have been applied to reveal candidate compounds that have the potential to become radiopharmaceuticals targeting DDR kinases [17].

## 2. Targeting DDR Pathways in GB

Under physiological conditions, the DDR protects the human genome by removing errors and avoiding the insurgence of mutations. However, in tumors treated with DNA damaging agents, DNA repair systems contribute to treatment failure [10,13,18,19]. Depending on the type of damage and cell cycle phase of the tumor cells, different actors of the DDR pathway are activated; these aspects have been previously reviewed [10,13,20]. Most of the subtle changes to DNA, such as oxidative lesions and single-strand breaks (SSBs), are repaired through different pathways such as base excision repair (BER), nucleotide excision repair (NER), or mismatch repair (MMR) [19]. SSB repair is activated following TMZ chemotherapy [21]. DNA double-strand breaks (DSBs), induced by RT, appear to be primarily repaired by non-homologous end-joining (NHEJ) or the error-free homologous repair (HR) pathway [19]. Surveillant sensor kinases in the DDR pathways, e.g., ATM, ATR, poly (ADP-ribose) polymerase-1 and -2 (PARP1 and PARP2) or the DNA-PK catalytic subunit (DNA-PKcs), recognize DNA damage and are recruited to the sites of SSBs and DSBs (Figure 2). Moreover, sensor kinases are responsible for the formation of protein complexes such as the MRN complex (Nbs1/hMre11/hRad50) or DNA-PK (Ku70/Ku80/DNA-PKcs). The DNA-PK and MRN complexes assemble and compete at sites of DNA DSBs where they act as damage sensors and initiate cell cycle dependent NHEJ or HR, respectively [22]. After sensor proteins detect DNA damage, transducer proteins with kinase activity, e.g., checkpoint kinase-1 (CHK1) and -2 (CHK2), trigger the activity of downstream effectors that can influence and/or direct a variety of cellular responses, including transcription processes, cell cycle regulation, DNA repair processes and apoptosis initiation [13].

Aberrant activation of these DDR kinases (ATM, ATR, DNA-PK, CHK1, CHK2, and PARP) in cancer is strongly correlated with resistance to genotoxic cancer therapies, including in GB [10,23]. Defects in the ATM-CHK2-p53 pathway promote GB formation and play a role in the response of glioma to ionizing radiation (IR) [24]. Mutations in isocitrate dehydrogenase 1 (IDH1) are frequently found in gliomas and are associated with better therapeutic outcomes. Interestingly, co-mutations in DDR kinases could play a role. In IDH1 mutated astrocytoma patients, TP53 (63%) and ATRX (27%) are the top two genes that display a higher frequency of mutations. An association between IDH1 mutations and reduced ATRX expression has also been shown. Mutations in CHK2 are instead associated with an IDH1-wildtype astrocytoma [25]. Núñez et al. discovered that mutant IDH1 helps maintain genomic stability in tumors by enhancing the DDR [26]. Glioma stem cells (GSC) have been shown to promote radioresistance by preferential activation of the DDR pathway through increased cell cycle checkpoint activation. This contributes to an increased DNA repair capacity and results in greater survival [9]. Since the standard therapy of GB includes TMZ and RT, which both aim to damage the DNA, DDR inhibition is being explored as a way to increase treatment efficacy [10]. For example, the inhibition of phosphatase and tensin homolog (PTEN) phosphorylation at Y240 sensitizes GB to IR by preventing enhanced DNA repair [27]. The standard TMZ chemotherapy modifies DNA or RNA at N^7^-guanine, O^6^-guanine, and N^3^-adenine by the addition of methyl groups. Methylated O^6^-guanine sites are usually repaired by MGMT. Since MGMT is often upregulated in GB, TMZ-resistance may occur. However, as TMZ also introduces N-site alkylations, which are normally repaired by BER, GB cells can be sensitized to TMZ by inhibiting key proteins of the BER pathway (such as PARP) [28]. A number of DDRi have already reached the clinic for the treatment of GB, including the PARP inhibitors (PARPi) olaparib and veliparib (Figure 3) [11,29].

Sensitivity towards DDRi is dependent on specific biomarkers, and the identification of treatment-responsive patients constitutes one of the key challenges associated with the clinical use of DDRi. Recent work has identified genomic and functional DNA repair assays that provide the identification of predictive and pharmacodynamic DDRi biomarkers (Figure 4) [30]. Mutational signatures associated with robust HR deficiency (HRD) primarily include alterations affecting BRCA1, BRCA2, PALB2, and two canonical RAD51 paralog genes (RAD51B, RAD51C). A more complex “BRCAness” signature has been defined to denote HRD tumors that share molecular features of BRCA1/2-mutant tumors, which are likely to benefit from DDRi [31,32,33]. The most promising biomarkers of BRCAness in GB relate to IDH1/2, epidermal growth factor receptor (EGFR), PTEN, MYC proto-oncogene, and estrogen receptors beta (ERβ) signatures [34]. For example, the anti-tumor effect of TMZ with ATMi or PARPi is enhanced in IDH1 mutant gliomas, and TMZ increases ATRi sensitivity in MGMT-deficient GB cells [35,36,37].

Strategies to define the mutational status of these genes include immunohistochemistry (IHC) and next-generation sequencing techniques. The requirement of whole-genome or whole-exome sequencing for the identification of selected gene signatures limits its widespread clinical utilization as a biomarker. However, new computational tools such as signature multivariate analysis and combinations of genomic analyses with single-cell imaging may increase the number of patients to be considered for treatments targeting HRD [38,39].

## 3. DDR (Radio)Pharmaceuticals

Single-photon emission computed tomography (SPECT) and positron emission tomography (PET) imaging could be utilized to identify patients that would benefit from DDRi therapy. Radiopharmaceuticals targeting DDR kinases have potential for the assessment of DDR target engagement (e.g., PARP activity) and may assist in monitoring response to DDRi or other DNA damaging treatments [40]. In addition, given the well-understood involvement of DDR during tumorigenesis, the ability to monitor these repair processes using PET or SPECT may facilitate the detection of earlier stages of carcinogenesis [41].

Upon confirming the expression of the DDR kinase in the GB tumor, therapeutic DDR targeting radiopharmaceuticals could be administered prior to or in combination with other DNA-damaging agents (e.g., TMZ and external beam RT), ultimately causing a synergistic anti-tumor response. Noteworthy, DDRi induced toxicity to healthy tissue might be limited due to intact DDR pathways in healthy cells (Figure 1) [42]. Interestingly, TRT agents targeting DDR kinases offer increased cytotoxicity compared to cold DDRi due to the additional radiation-induced DNA damage. Our group recently published a perspective on TRT for the treatment in GB, with a special focus on radiopharmaceutical requirements, including target and radionuclide selection, blood–brain barrier (BBB) passage, toxicity, validation, and combined therapy strategies [43].

The nature of the induced DNA damage in TRT is dependent on the specific radiation characteristics of the used isotopes. Most GB research has investigated the cellular and physical effects of IR in the context of external beam RT, radiation effects which are significantly different compared to TRT radiation effects [20]. Lutetium-177, iodine-131, rhenium-186, rhenium-188, or yttrium-90, are commonly utilized for TRT of GB, featuring β^-^-particle emissions with relatively low linear energy transfer (LET) (0.2–2 keV/μm) and a low relative biological effectiveness (RBE). As a result, the β^-^-emission induced damage consists of some DSBs but mostly repairable SSBs, which could result in sublethal damage repair [44]. Targeted α-particle therapy (TAT) using astatine-211, actinium-225, or bismuth-213, is gaining attention due to the higher LET (50–230 keV/μm) and RBE inducing more complex DNA damage and a lower dependency on the tumor oxygenation status [45,46]. Complex DNA damage significantly contributes to exceeding the cellular capabilities of DNA repair, thereby forcing cells towards cell death [20]. The first positive clinical trials on TAT have emerged, and TAT was suggested as a facilitator to overcome tumoral resistance to chemotherapy [47,48]. A nice example is the astatine-211 radiolabeled PARPi, which induced cellular lethality by targeting alpha-emitters directly to the nucleus, with high sensitivity in neuroblastoma in vitro and in vivo. The [^211^At]-PARPi was 10,000 times more potent than talazoparib, indicating that the likely mechanism of cell killing does not rely on pharmacological PARP inhibition but rather on alpha-particle induced DNA damage [49,50]. Lastly, the short penetration range and LET (4–26 keV/μm) of Auger electron emitters make them suitable candidates for inducing damage to a specific target with dimensions comparable to the DNA, leading to complex, lethal DNA damage [51].

A current list of radiopharmaceuticals targeting DDR processes in various cancer types is provided in Appendix A and summarized in Figure 5. So far, most DDRi have been directly radiolabeled with ^123^I-, ^131^I-, ^18^F- and ^211^At-radionuclides. Only one analog of olaparib was ^64^Cu-radiolabeled following conjugation of a 1,4,7,10-tetraazacyclododecane-1,4,7,10-tetraacetic acid (DOTA) moiety, allowing for PET imaging of mesothelioma [29,40,49,50,52,53,54,55,56,57,58,59,60,61,62,63,64,65,66,67,68,69,70,71,72,73,74,75,76,77,78,79,80,81,82,83,84,85,86,87,88,89,90,91].

Interestingly, it was also hypothesized that ‘cold’ DDRi could increase the effectiveness of TRT agents. This hypothesis has already been confirmed in ovarian cancer xenografts, where the synergistic effect of a mesothelin-targeted ^227^Th-conjugate in combination with ATMi, ATRi, DNA-PKi, and PARPi was investigated [92]. In prostate and neuroendocrine cancer, multiple clinical trials are still running, combining PARPi with [^177^Lu]-DOTATATE, [^177^Lu]-PSMA-617 or [^223^Ra]-dichloride (ClinicalTrials.gov Identifiers: NCT03874884, NCT03317392, NCT03076203, NCT04086485, NCT04375267) [29]. Unfortunately, studies have not been initiated in GB yet.

### 3.1. ATM/ATR Inhibitors

#### 3.1.1. Targeting ATM/ATR as an Anti-GB Strategy

ATM and ATR are members of the PIKK family of serine/threonine protein kinases, which are crucial in the initiation of cell cycle arrest and apoptosis (Figure 1). ATM is the main kinase in the cellular response to DNA DSBs, while ATR is activated by single-stranded DNA structures, which may arise upon SSB induction and stalled or collapsed replication forks. Although ATM and ATR are activated by different types of DNA damage, their signaling cascades are partially overlapping [93,94,95]. For example, CHK1/2 is a downstream target in both pathways. However, ATM plays a crucial role in the activation of the G1/S cell cycle checkpoint while ATR enforces the intra-S-phase and G2/M cell cycle checkpoint (Figure 1) [96]. Notably, the list of substrates undergoing ATM-dependent phosphorylation is still growing [97].

Hypersensitivity of ATM-defective cells to IR and the critical function of the ATR pathway for the survival of tumor cells has led to considerable interest in ATM and ATR as therapeutic targets for cancer therapy [93,98,99]. Glioma cells, especially GSCs, exhibit increased resistance to IR, which is mediated by an upregulation of DDR targets such as ATM, ATR, PARP1, and CHK1. This results in a rapid G2/M cell cycle checkpoint activation and enhanced DNA repair [9,100]. However, tumor cells often suffer from defects in ATM function through mutation of the ATM protein itself or its associated downstream targets, particularly p53. Such mutated cells must maintain functional S and G2/M cell cycle checkpoints mediated by ATR/CHK1 to avoid premature mitotic entry [101]. The genomic characterization of human GB genes and their core pathways showed that p53 signaling was altered in 87% of GB cases [102]. Therefore, ATR/CHK1 inhibition shows great potential to induce synthetic lethality [12,102,103]. Treatment with ATM- or ATR-inhibitors (ATMi/ATRi) may thus selectively sensitize glioma cells and GSCs to IR and/or TMZ [103,104,105].

Possible determinants of ATRi sensitivity include high levels of ATR, Cdc25A, and CHK1. Multiple predictive biomarkers have also been incorporated into early phase trials: alternative lengthening of telomeres, reduced expression/loss of function of ATM, BRCA1/2, TP53, ARID1A, and overexpression of CCNE1, APOBEC, and MYC (Appendix A and Figure 4) [106,107,108]. Especially DDRi combined with IR could provide a therapeutic strategy for IDH1^R132H^ glioma patients who also harbor p53- and ATRX-inactivating mutations [26]. Alpha Thalassemia/Mental Retardation Syndrome X-Linked (ATRX)-deficient glioma displaying p53 loss of function could also benefit from ATRi therapy [109,110]. In p53-deficient settings, suppression of ATM dramatically sensitized cells to chemotherapy, whereas, conversely, ATM suppression had the opposite effect in the presence of functional p53 [101]. ATM kinase inhibition combined with low dose radiation was also selectively toxic to glioma with mutant p53 through the induction of mitotic catastrophe and apoptosis [111]. Overexpression of cMYC has previously been shown to cause replicative stress and to confer sensitivity to CHK1i and ATR knockdown. Mutations in ARID1A predict response to ATRi and PARPi since ARID1A-deficient cells rely on ATR checkpoint activity to prevent apoptosis [112]. Lastly, pRAD50 has been identified as a novel and clinically applicable pharmacodynamic biomarker of sensitivity to ATM/ATR inhibition [113].

#### 3.1.2. Current Status of ATM/ATR Targeted Therapy in GB

An overview of oncological clinical trials investigating ATMi and ATRi and their relevant biomarker selection criteria are summarized in Table 1 and Appendix A. There are currently four ATMi (KU-60019, AZD0156, AZD1390, and M3541) evaluated in clinical trials, of which two (AZD1390 and AZD0156) include glioma patients (Table 1) [103]. One of the first-generation ATMi includes the small molecule ATP-competitive inhibitor 2-morpholin-4-yl-6-thianthren-1-yl-pyran-4-one (KU55933) and its ameliorated derivatives KU-60019 and CP466722 [114,115,116]. Therapy with these ATMi has resulted in chemo- and radiosensitization of GB cells and a significant two- to three-fold increased survival when KU-60019 was administered intratumorally in GB models combined with external beam IR. Particularly, a signature of IDH mutations combined with a low expression of TP53 or MGMT and high expression of phosphatidylinositol-3-kinase (PI3K) has been identified as a biomarker for more effective ATM-based therapy [35,117,118,119,120,121]. Interestingly, KU-60019 limited glioma cell growth in co-culture with human astrocytes, with the latter seemingly unaffected by the same treatment [104,115,118,120]. The last generation ATMi AZ32 and AZD1390 have been specifically designed to effectively cross the BBB and showed radiosensitizing effects in GB both in vitro and in vivo [75,93,103,111]. This led to a phase I study of AZD1390 in combination with RT in patients with brain cancer (ClinicalTrials.gov Identifier: NCT03423628). The ATMi AZD0156 has shown potential in multiple cancer types, including synergism with PARPi [29,122,123,124]. In a phase I trial combining AZD0156 with olaparib, hematologic toxicity appears to be the treatment-limiting toxicity in advanced malignancies (including glioma), although efficient doses could be reached [125]. Data on ATMi KU-59403 in GB are awaited [126]. Finally, besides employing small molecule ATMi, silencing of ATM or ATR using siRNA has also been shown to increase glioma cell chemo- and radiosensitivity [21,127,128].

ATRi has demonstrated significant therapeutic potential in cancer treatment, with anti-tumor activity when administered as monotherapy but also when combined with conventional chemotherapy, RT, or immunotherapy [99]. The ATRi currently in clinical trials are VX-970 (also known as VE-822, M6620, or berzosertib, Merck^®^, Darmstadt, Germany), VX-803 (M4344, Merck^®^), BAY1895344 (elimusertib, Bayer^®^, Leverkusen, Germany), M1774, RP-3500, and AZD6738 (ceralasertib, AstraZeneca^®^, Cambridge, UK). Notably, some of these trials considered biomarkers for patient stratification (Appendix A). Unfortunately, no trials have been initiated in GB so far [29,107,133,134].

VX-970, for which 15 trials are now active, reached the clinic first [29,124,135,136]. Radiation and chemosensitization effects have been shown, but efflux pump mechanisms limit brain accumulation of VX-970 [37,137,138,139]. However, prolonged survival was noted in rats with intracranial GB tumors that were treated with RT combined with VX-970. Survival was even more improved upon the combination with PARPi [140,141]. The synthetic lethal interaction of VX-970 might be enhanced by selecting another ATR/CHK1 downstream target, such as WEE1. WEE1 inhibitors (WEE1i) have recently attracted attention with multiple phase I/II studies investigating this synergy (Figure 1) [29,142,143]. WEE1 promotes S and G2/M cell cycle arrest by blocking cyclin-dependent kinase 1 and 2 (CDK1/2) and allowing DNA repair, as shown in Figure 2 [144]. The most studied WEE1i is adavosertib (MK1775, AZD1175), with 23 active trials, including a phase I trial in GB patients (ClinicalTrials.gov Identifier: NCT01849146) [29]. In addition, 27 clinical trials are currently actively evaluating the selective ATRi AZD6738 (Appendix A) after promising preclinical results [29,99,145,146]. Notably, no significant radiosensitizing effect was found in an orthotopic GB animal model despite effective AZD6738 brain penetration [147].

NVP-BEZ235, a dual PI3K/mammalian target of rapamycin (mTOR) inhibitor, was identified as a potent inhibitor of ATR and ATR homologs, ATM and DNA-PK [148]. NVP-BEZ235 effectively crosses the BBB with GB radiosensitization and TMZ sensitization effects, but toxicity was shown upon introduction in the clinic [29,147,149,150,151,152,153,154,155,156,157]. Several ATRi have been abandoned in the development stage before reaching the clinic, including Schisandrin B, NU6027, ETP-46464, VE-821 (later optimized to VE-822/VX-970), and AZ20 (later optimized to AZD6738) [99].

#### 3.1.3. ATM/ATR Radiopharmaceuticals

Two ATMi have been ^11^C-radiolabelled: AZD1390 and AZD0156. In macaque monkeys, intravenous administration revealed superior permeability and BBB penetrating properties of [^11^C]-AZD1390 compared to [^11^C]-AZD0156 [75]. A first clinical trial in healthy volunteers analyzed the brain distribution of [^11^C]-AZD1390 and confirmed BBB penetration [59]. These findings support the use of radiolabeled AZD1390 for therapy and/or diagnostics in patients with central nervous system (CNS) malignancies, including GB. Only one ATRi, VE-821, a less potent precursor of the ATRi VE-822, has been ^18^F-radiolabelled. This VE-821 analog (termed ‘[^18^F]-ATRi’) was put forward as a clinically relevant PET imaging agent in an in vivo study by Carlucci et al., and specific target binding was confirmed using a U251 MG GB animal model [40].

### 3.2. CHK1/2 Inhibitors

#### 3.2.1. Current Status of CHK1/2 Targeted Therapy in GB

CHK1/2 are cell cycle checkpoint kinases that prevent cell cycle progression when DNA damage is detected and being repaired, as shown in Figure 2 [158,159].CHK1 is activated by ATR phosphorylation on Ser317 and Ser345, and CHK2 is activated by ATM phosphorylation on Thr68. CHK2 phosphorylates p53, preventing its interaction with MDM2, and subsequently, p53 drives the expression of genes involved in apoptosis induction and cell cycle checkpoint activation, such as p21/CDKN1 [96]. CHK1 plays an important role in intra-S-phase and G2/M cell cycle checkpoint progression mediated by phosphorylation and inhibition of Cdc25A and Cdc25C [93,159]. Inhibited Cdc25 proteins are no longer able to activate their CDK proteins substrates and thereby fail to induce cell cycle arrest [160].

CHK1/2 upregulation has been shown in GB, and inhibition is of interest, particularly in GBs with aberrations in other cell cycle regulating factors, such as p53, since these tumors rely on the remaining checkpoints to repair DNA damage. Approximately 50% of GB patients with CHK2 alterations also carry defects in the p53 signaling pathway, while this is only 10–13% for DDR components ATM, ATR, or CHK1 [9,24,161]. In GSCs, the basal expression of CHK1 and Cdc25C has also shown to be much higher compared to differentiated GB cells [100,161].

CHK1/2 inhibition has been extensively explored clinically in various cancer types but not yet in GB, likely because numerous CHK1/2i were discontinued before phase III, such as UCN-01 (7-hydroxystaurosporine), rabusertib (LY2603618) and MK-8776 (SCH 900776) [162,163,164,165,166,167,168]. AZD7762, for instance, showed severe cardiac toxicities in patients with advanced solid tumors (AST) [169]. Clinical trials are currently ongoing for CHK1-selective inhibitors CCT245737 (SRA737), GDC0575 (ARRY-575, RG7741), and the CDK1/2 inhibitor prexasertib (LY2606368). Prexasertib-related neutropenia has been identified as an adverse effect but warrants further development with clinical activity in ovarian cancer, squamous cell carcinoma, and advanced cancer types [170,171,172,173,174]. The CHK1i GDC-0425 or GDC-0575, given in combination with gemcitabine to solid tumor patients, both warrant further investigation [175,176].

CHK1i therapy of GB has remained in the preclinical setting. Treatment with gemcitabine and the CHK1i MK-8776 effectively permeated the BBB and inhibited glioma growth in vivo [177]. Moreover, UCN-01, although in itself non-toxic, increased the cytotoxicity of TMZ by five-fold in U87MG (p53 wild-type or deficient) glioma cells by accumulating the number of cells bypassing G2-M arrest and thereby undergoing mitotic catastrophe [178]. UCN-01 also inhibited GSC growth in vitro, and AZD7762 radiosensitized p53-mutated GB cell lines (confirmed in GB in vivo models) [179,180,181]. SAR-020106 sensitized human GB cells to RT, TMZ, and decitabine treatment [182]. The impact of CHK1 inhibition on GB cells was also studied using SB18078 and PF477736, confirming an influence on colony and tumor sphere formation, as well as cell proliferation. Khanna et al. also confirmed that CHK1 acts via protein phosphatase 2A in promoting GB cell growth [183]. Unfortunately, in AST, a phase I study on PF477736 combined with gemcitabine was terminated due to business reasons (NCT00437203) [29]. Interestingly, targeting the CHK1 gene in GSCs, using, for example, lentivirus-delivered short hairpin RNA (shRNA), also showed the potential to increase radiosensitivity via apoptosis induction [184].

Less research is performed on CHK2i in GB. It should be noted that while knockdown of CHK1 expression enhanced radiosensitivity of human GSCs, this was not the case upon CHK2 inhibition [184]. TMZ-induced cell death was also more prominently enhanced by pharmacologic inhibition of CHK1 compared to CHK2 inhibition [128]. However, the CHK2i PV1019 radiosensitized U251 glioma cells [12,185]. As an alternative to CHK1/2 inhibition, inhibition of their downstream targets CDK1/2 or Cdc25A protein phosphatase has been studied [186]. In our opinion, the multi-targeted MAPK inhibitor MEK162, which also inhibits CDK1/CDK2/WEE1/p-ATM besides CHK2, should be further explored since it downregulated and radiosensitized spheroidal and orthotopic GB xenografts [15].

#### 3.2.2. CHK1/2 Radiopharmaceuticals

Therapeutic effects of CHK1/2 inhibition can be visualized using molecular imaging techniques such as PET or MRI. For example, radiosensitization effects after CHK1/2i therapy were visualized using diffusion-weighted MRI in GB models [181]. The proliferation PET tracer 3′-deoxy-3′-[^18^F]fluoro-thymidine ([^18^F]FLT) was also able to visualize antiproliferative effects in xenograft rodents following PF00477736 therapy [187]. Unfortunately, radiolabeled CHK1/2i are scarce. Both CHK1/2i prexasertib and CHK1i LY2603618 (rabusertib) were radiolabeled with carbon-14 to study their metabolism in advanced/metastatic solid tumor patients [60,91]. Also, [^14^C]-GDC-0425 was used to evaluate safety concerns of thiocyanate arising from GDC-0425 administration, but these proved to be negligible [90].

### 3.3. PARP Inhibitors

#### 3.3.1. Current Status of PARP Targeted Therapy in GB

PARPi have shown significant promise in a variety of malignancies with deficiencies in HR signaling [34,188]. In GB, the BRCAness phenotype leads to impairment of HR and thus PARPi sensitivity [34]. Glioma biomarkers of predictive value for PARPi therapeutic efficacy include IDH1/2 mutations, a low BRCA1 expression, aberrant ATM or ATR signaling, MYC overexpression, and inactivation of mismatch repair genes, especially MSH6 [36,123,189,190,191,192,193,194]. PTEN mutations, present in 70% of GB tumors, have shown to increase the level of DSBs upon PARP inhibition, though some studies contradict this [195,196,197,198]. MGMT promoter hyper methylation is also being studied as a potentially predictive biomarker for PARPi-mediated TMZ sensitization [189]. TMZ-induced damage can be repaired by either direct repair (in case of O^6^-methylguanine lesions) or BER (in case of N^7^-methylguanine and N^3^-methyladenine lesions). Thus, inhibiting PARP-mediated SSB repair (BER) leads to the accumulation of DNA DSBs, thereby enhancing cytotoxicity [199,200]. This way, glioma patients may still benefit from alkylating chemotherapy, regardless of their MGMT promotor status [200,201,202]. Another mechanism of PARPi TMZ sensitization is allosteric PARP trapping (leading to instability of stalled replication forks), as well as BRCA1 and RAD51 depletion (leading to compromised fork protection) [189,203]. For more info on the combination effects of PARPi and chemotherapeutics, we refer the reader to [204]. Interestingly, cancers with BRCA-deficiency and PARPi resistance could also benefit from a combined therapy including CHKi and PARPi [174,205,206]. CHK2 inhibition might also provide a strategy to alleviate hematologic toxicity from PARPi [207].

Currently, four PARPi have been FDA-approved: olaparib (AZD2281 or KU0059436, Lynparza^®^, AstraZeneca); rucaparib (Rubraca^®^, Clovis Oncology, Boulder, CO, USA); talazoparib (Talzenna^®^, Pfizer, Manhattan, NY, USA); and lastly, niraparib (Zejula^®^, Tesaro, Waltham, MA, USA). A fifth PARPi, veliparib (ABT-888, Abbott Laboratories, Abbott Park, IL, USA), is expected to obtain approval in the near future, following promising phase III trial results in metastatic breast cancer [29,188]. In-depth reviews of the current status of PARPi as mono- or combination therapies for cancer were previously published [162,188,208].

Preclinically, olaparib delayed GB recurrence when combined with RT and sensitized IDH1-mutated tumor cells when combined with TMZ, leading to clinical trials in GB patients (Table 1) [29,36,209,210]. The phase I OPARATIC trial in recurrent GB patients confirmed that olaparib could be safely combined with daily TMZ if intermittent dosing was applied. Additionally, drug penetration into the entire tumor specimen was confirmed [129]. A phase I/II study in GB of olaparib combined TMZ/RT is currently recruiting [29,211,212].

The radiosensitizing effect of the PARPi veliparib (ABT-888) has been shown preclinically in GB, despite two studies proving otherwise [195,200,210,213,214,215,216,217,218,219,220]. The addition of veliparib to TMZ also prevented TMZ resistance, although this may not be achievable in a tolerable dosing regimen [191,221,222,223]. In recurrent GB previously treated with bevacizumab, the TMZ/veliparib combination did not significantly improve six-month progression-free survival [131,224]. Administering veliparib in combination with standard RT/TMZ was also not tolerable in GB patients and did not provide clinical benefit in unmethylated MGMT GB patients [130]. The MGMT-methylated GB patient population will be addressed in the Alliance A071102 trial (ClinicalTrials.gov Identifier: NCT02152982) [29]. Inhibition of ABCB1 and ABCG2 (drug efflux transporters expressed at the BBB) by elacridar may improve the efficacy of TMZ/veliparib therapy [196]. Combined veliparib/RT/TMZ is also being explored in malignant glioma patients without H3 K27M or BRAFV600 mutations (Table 1) [29].

Rucaparib has shown anti-GB effects in vitro, which were ameliorated when combined with BKM120 (PI3K inhibitor) or when conjugated to IR-786 (heptamethine cyanine dye) [225,226]. In combination with TMZ, rucaparib prolonged the time to tumor regrowth by 40% in heterotopic GB xenografts. However, this could not be confirmed in orthotopic GB models, most likely due to limited drug delivery [227]. Despite being FDA-approved for various cancer types, rucaparib has not yet been investigated in clinical trials for GB patients. In AST, however, rucaparib/TMZ was well-tolerated and showed proof-of-principle [29,228].

The PARPi talazoparib is FDA-approved for breast cancer, and a phase II trial on the talazoparib/carboplatin combination is currently recruiting recurrent high-grade glioma patients with DDR deficiency [29]. The combination of high and low LET radiation qualities with talazoparib led to promising preclinical results when administered to GSCs. Moreover, EFGR amplification might increase their sensitivity [229,230,231]. In vivo, talazoparib combined with TMZ prolonged GB stasis, but this could not be confirmed in orthotopic GB models, most likely due to BBB efflux mechanisms [232].

Niraparib (MK-4827) is currently being investigated in recurrent GB, either combined with RT or with tumor-treating fields (TTFs). TTFs are expected to reduce BRCA1 signaling and thereby reduce DNA repair capacity, causing PARPi-assisted synthetic lethality [29]. The first results on the niraparib/TMZ combination indicated tolerance and efficiency in patients with advanced cancer [132]. Notably, niraparib penetrated intracranial tumors in breast cancer models [233].

Other PARPi under investigation in GB include pamiparib (BGB-290, Partruvix™; BeiGene Ltd., Changping Qu, Beijing, China), E7016 and CEP-8983 (prodrug CEP-9722) [215,234,235]. Preclinically, pamiparib has shown strong anti-tumor synergism with TMZ and improved BBB penetration compared to other PARPi, which led to clinical trials, as outlined in Table 1 [29,197]. In a phase I trial in patients with solid tumors, CEP-9722 showed limited clinical activity [236].

Finally, it was shown that combined inhibition of PARPi and ATRi in GSCs resulted in a profound radiosensitization, which exceeded the effect of a single ATRi [100,141]. Multiple clinical trials are exploring this combination (olaparib/ceralasertib), including a phase II trial in IDH mutant solid tumors (ClinicalTrials.gov Identifier: NCT03878095) [29].

#### 3.3.2. PARP Radiopharmaceuticals

Radiolabeled versions of PARPi strongly gained momentum in the last years due to their potential to directly and non-invasively image PARP expression, quantify the biodistribution of a PARPi and its tumor uptake, define treatment response and stratify patients likely to respond to PARPi therapy [199]. Due to the nuclear sub-cellular location of PARP and confirmed overexpression in GB, with overall low expression in healthy brain tissue, PARP-1 is also a near-ideal target to develop radiotherapeutics [56,188]. In addition to eliciting synthetic lethality, promoting genomic instability, and enhancing cytotoxicity of a subsequently administered DNA-damaging agent, PARP-TRT could also cause DNA damage [237]. In response to DNA damage, the expression of PARP-1 also increases, which may result in an increased target availability for the therapeutic radiopharmaceutical [50].

Most radiopharmaceuticals targeting PARP are structurally similar to small molecules olaparib ([^18^F]-BO, [^18^F]-PARPi, [^18^F]-20, [^123^I/^131^I]-PARPi, [^18^F]-PARPi-FL, [^64^Cu]-PARPi, [^18^F]-olaparib, [^18^F]-AZD2461, [^18^F]-AZD2281, [^11^C]-PJ34) or rucaparib ([^18^F]FTT), [^18^F]-WC-DZ-F, [^18^F]FE-LS-75, [^125^I]-KX1, [^125^I]-KX-02–019, [^14^C]-rucaparib, [^211^At]-MM4) [40,50,53,56,64,69,78,87,188]. These radiopharmaceuticals were recently reviewed [188,237,238]. Three of these PARP radiopharmaceuticals, namely [^18^F]-PARPi, [^18^F]FTT, and [^14^C]-rucaparib have reached the clinical setting (Appendix A) [61,69,76,85].

[^18^F]-PARPi, was deemed well tolerable and safe in patients with head-and-neck cancer [61]. In GB mouse models, [^18^F]-PARPi and a bimodal fluorescence/PET imaging agent succeeded in visualizing the tumor [40,53,54]. Additionally, [^18^F]-PARPi has shown potential in discriminating active brain cancer from treatment-related changes in a murine model of radiation necrosis. This was confirmed in brain cancer patients, including three patients with IDH wild-type primary GB [76,77]. [^18^F]-PARPi-PET/MRI is currently being evaluated in a pilot study in recurrent brain tumors (ClinicalTrials.gov Identifier: NCT04173104) [29]. [^18^F]FTT is currently being investigated in phase I studies in various cancer types, including GB (Appendix A) [29,85]. [^18^F]FTT-PET was, for example, performed to measure PARP-1 expression pre- and post-treatment with TTF and niraparib. Additionally, [^18^F]FTT uptake was correlated with HR deficiency status [29]. Unfortunately, early clinical results of [^18^F]FTT report low brain penetration and high uptake values in the liver and spleen [86]. In addition, regrettable results have also been reported for other olaparib-based radiopharmaceuticals. An absorption, metabolism, and excretion (ADME) analysis of [^14^C]-rucaparib, reported no brain uptake, and development of [^18^F]-20 was halted due to substantial defluorination [53,69].

PARP radiopharmaceuticals in a preclinical phase that are worth exploring in GB include [^14^C]-pamiparib and [^64^Cu]-DOTA-PARPi. The ADME of [^14^C]-pamiparib was evaluated in four patients with advanced cancer and indicated near-complete absorption and low renal clearance of the parent drug [74]. [^64^Cu]-DOTA-PARPi showed potential in mesothelioma-bearing animal models [64]. Notably, fluorinated radiopharmaceuticals based on talazoparib have been evaluated in a prostate cancer model and indicated TRT potential [65].

Therapeutic radiopharmaceuticals targeting PARP have been studied in GB preclinically with promising results. The first Auger-based theranostic PARPi, the Iodine-123 Meitner-Auger PARP1 inhibitor, successfully delivered a lethal payload within a 50 Å distance of the DNA of GB cancer cells and demonstrated a survival benefit in mouse models of GB [57,239,240]. [^123^I] -I2-PARPi retained within GB xenograft tumors and correlated with PARP expression [55]. Jannetti et al. developed [^131^I] -PARPi, a 1(2H)-phthalazinone with a similar structure to olaparib. Convection enhanced delivery of [^131^I] -PARPi led to increased survival of mice with orthotopic brain tumors [56]. Selective binding of ^131^I- and ^124^I-labelled I2-PARPi was also confirmed in GB models [58]. A particularly promising PARP-TRT agent is [^211^At]-MM4, a rucaparib derivative, due to its high cytotoxicity and favorable half-life (7.2 h). In neuroblastoma models, this compound resulted in increased survival [50].

### 3.4. DNA-PK Inhibitors

#### 3.4.1. Current Status of DNA-PK Targeted Therapy in GB

DNA-PK consists of a heterodimer (Ku70/80) and a large catalytic subunit, known as DNA-PKcs [12,241]. This complex initiates NHEJ by binding to the DSB, leading to subsequent phosphorylation and activation of DNA-binding proteins, ultimately causing ligation of DSB ends (Figure 1) [10]. In GB, high DNA-PK levels correlate with poor survival and increased GSC stability [242,243]. DNA-PK has been shown to mediate GSC radioresistance and glioma progression in vivo, suggesting DNA-PK/RAD50 as promising targets for GSC eradication [244]. To date, research on biomarkers demonstrating DNA-PK inhibition sensitivity is only preclinical, but HR deficiency could theoretically predict sensitivity to DNA-PKi given the increased reliance of HR deficient cells on NHEJ [162]. Sun et al. identified p53 as a potential predictive biomarker of response to the combination of DNA-PKi and RT [245].

Small molecule inhibitors of DNA-PK, from the discovery of the first identified inhibitors (wortmannin and its derivatives PX-866 and PWT-458) to more selective DNA-PKi, have been reviewed [13,246,247]. The DNA-PKi VX-984 (Vertex, now licensed to Merck KGaA, Darmstadt, Germany as M9831), nedisertib (M3814, peposertib, MSC2490484A, Merck KGaA), and the recently discovered AZD7648, have entered clinical trials. Only nedisertib combined with RT/TMZ is currently under investigation for GB patients with unmethylated MGMT promotor status, following preclinical evidence of a radiosensitizing upon NHEJ inhibition [29,248]. In phase I trials for AST, this combination was well tolerated and demonstrated modest efficacy [249]. VX-984 has shown promising radiosensitizing effects in GB in vitro and in vivo with confirmed BBB crossing [250]. Interestingly, an inability to resolve γ-H2AX foci in the presence of VX-984 could be induced in T98G cells only [251]. Results on the safety of VX-984 administered in AST patients are still pending [29]. AZD7648 is undergoing clinical evaluation in AST after it showed RT/TMZ sensitizing effects and synergism with olaparib. However, this needs to be confirmed in GB [29,252,253].

Less selective DNA-PKi co-targeting mTOR include CC-115, avadomide (CC-122), samotolisib (LY3023414), and NVP-BEZ235. In GB patients, CC-115 was well tolerated, and 21% achieved a stable disease status with proven drug distribution in GB tissue [254,255]. CC-115 has shown a synergistically lethal effect with functional ATM loss and is included in one of the three experimental arms of the ongoing Individualized Screening Trial of Innovative GB Therapy (INSIGhT) trial [254,256]. Avadomide (CC-122) has recently been deemed safe in various cancer types, and applicability in CNS-related cancers was suggested. A phase I trial on avadomide in patients with advanced tumors unresponsive to standard therapies, including GB, is still active (ClinicalTrials.gov Identifier: NCT01421524) [257]. Samotolisib (LY3023414) had single-agent activity in advanced cancer patients and is being investigated further in pediatric CNS tumors. To be noted, BBB penetration remains a stumbling block [29,258,259,260].

Finally, based on preclinical data alone, SU11752, KU0060648, NU7026, and NU7441 have been put forward as glioma targeted drugs, either as single agents or in combination regimens (RT/TMZ/topoisomerase II inhibitors) [261,262,263,264,265].

#### 3.4.2. DNA-PK Radiopharmaceuticals

Radiopharmaceuticals targeting DNA-PK are scarce. In healthy subjects, the disposition of the samotolisib derivative [^14^C]-LY3023414 following oral administration was studied; however, results are pending (ClinicalTrials.gov Identifier: NCT02575703) [29]. It should be noted that the uptake of radiolabeled LY3023414 would not be DNA-PK specific because it also targets PI3K/mTOR. Additionally, the radiosynthesis protocols for ^11^C-labelled chromen-4 derivatives as new potential DNA-PK-PET imaging radiopharmaceuticals were published by Gao et al. but have not yet been validated in vivo [73].

## 4. Development of Other DDR Radiopharmaceuticals

Besides DDRi radiopharmaceuticals themselves, radiotracers enabling the visualization and quantification of the amount of DNA damage induced after TRT would be extremely valuable, e.g., to assess the radiobiological treatment response of the tumor. This category includes γH2AX radiotracers: ^89^Zr-/^111^In-labelled anti-γH2AX-TAT. Anti-γH2AX antibodies are routinely used in ex vivo assays to quantify the number of γH2AX foci or DNA DSBs within cell populations, but a cell-penetrating peptide is required for in vivo applications [41,266]. For example, the extent of DNA damage response after [^177^Lu]-DOTATATE therapy was evaluated using [^111^In]-anti-γH2AX-TAT SPECT imaging [267].

## 5. Challenges and Risks of DDRi (Radio)Pharmaceuticals

Exploiting synthetic lethal interactions has attracted considerable attention as an anticancer strategy; however, the development of such approaches to selectively target cancer cells while sparing health tissues remains challenging [158]. Major hurdles include tumor biology, heterogeneity, and complexity; an inadequate understanding of synthetic lethal interactions; drug resistance, and the challenges regarding screening and clinical translation. Hence, there is an urgent need to develop improved efforts aiming to identify and understand synthetic lethal interactions, as well as validate new screening tools and biomarkers, including DDR radiopharmaceuticals. Improved genetic perturbation techniques, including CRISPR/Cas9 gene editing, are also promising prospects concerning synthetic lethal effects in cancer [268].

DDRi induced toxicity to healthy tissue can be limited due to the innate DDR pathways in healthy cells (Figure 1). The phenomenon of “replication stress”, unique to fast proliferating cancer cells, enforces this statement [16,42]. Unspecific cellular toxicity may occur since most DNA repair pathways overlap in terms of DNA repair proteins. This could lead to unwanted DNA damage to normal tissue, increasing the risk for late toxicity [158]. For example, ATM inhibition showed a greater radiosensitizing effect in p53-deficient tumors, but effects were also observed in p53 wild-type cells [111]. This might be an important consideration for proliferating cells of the CNS, where a p53-dependent G1/S checkpoint would stay at least partially activated in the presence of an ATMi (via ATR), thereby inducing cell cycle arrest and preventing apoptosis. In neurons, ATM seems to be a requirement for apoptosis. Hence, transient brain exposure to an ATMi might not be extremely toxic [111]. Upon PARP inhibition, toxicity to the normal brain is expected to be minimal since PARP-1 expression has not been detected in normal neurons [269]. Moreover, early-phase clinical trial data indicates that the radiosensitizing properties of PARPi are most pronounced in rapidly proliferating cells [212]. Hence, due to the non-dividing nature of neuronal tissue in the brain, it is assumed that the addition of a PARPi to RT would have a relatively larger effect on highly proliferative GB cells compared to normal brain cells [200]. The toxicity of PARPi is also related to their PARP trapping capacity, and reactivities differ with different combination partners and the DNA damage mutations present [204,270]. For example, the combination of PARPi with chemotherapy is hampered by overlapping toxicities, thereby limiting their administrable dose. Interestingly, hematologic toxicity seems more pronounced in germline BRCA carriers [270,271].

In the context of TRT, the combined toxicity of the cold DDRi with the radionuclide is important to consider. TRT toxicity can be related to targeting efficiency, radionuclide stability and the nuclear recoil effect, physical properties of the radionuclides, dosimetry, immunogenicity, and administration route [43]. Confirming the presence of the DDR target using an imaging DDR radiopharmaceutical (SPECT/PET) and evaluating its distribution throughout the body (before selecting TRT as a treatment strategy) is essential. Increased toxicity might be expected in case multiple DNA damaging strategies are combined. However, it should be noted that the concentration of the DDRi for targeted therapy will be markedly higher than the prospective dosage given of a radiolabeled DDRi during nuclear imaging or TRT. Following PARP-TRT, normal tissue toxicities in the spleen and bone marrow are projected due to PARP-1 expression in normal tissues. Other potential sites for toxicity include the liver and gastrointestinal tract if involved in the biological clearance of the compound [50].

Nuclear imaging strategies have shown the ability to measure expression levels of DDR kinases in vivo. However, when compared to the tumor uptake of radiopharmaceuticals targeting cancer biomarkers situated on the cell surface, uptake of these agents is generally low. Factors such as the transient nature of DDR protein activation (e.g., following RT, the expression levels of many biomarkers, including PARP-1 and γH2AX, disappear within days), the inefficient drug internalization/nuclear translocation, and specifically for GB applications, BBB crossing play a role [41]. In the case of alpha emitters, sub-cellular delivery to cell nuclei will increase the cytotoxicity due to the high probability that both the alpha particle and its atomic parent nuclei recoil radiation will traverse the cell nucleus [50]. As can be seen in Appendix A, most of the DDRi investigated for TRT have been radiolabeled with halogens. Reaching the nucleus might be difficult upon radiometal chelation; however, some preclinical results showed promise. A Cu-64-radiolabeled olaparib analog containing a DOTA moiety resulted in clear tumor uptake in mesothelioma [64]. The nuclear uptake of a [^177^Lu]-DOTA-labeled DNA intercalator in Raji cells was deemed sufficient, although it was lower compared to total cellular uptake [272].

Treatment resistance is a bothersome limitation for the application of DDRi and DDR-based TRT. GB tumors relying on one DNA repair pathway for their survival may additionally hold mutations that cause resistance to certain DDRi [158]. For example, PARPi resistance may be induced by HR restoration or mitigation of replication stress. Identified biomarkers for PARPi resistance include loss of 53BP1/ARID1A, low level of Schlafen 11 (SLFN11) or GBP1, and genomic reversion of BRCA1/2. In addition, DNA replication fork protection (PTIP/EZH2) or genetic mutations that result in the activation of a drug efflux pump play a role. This highlights the need for functional biomarkers that can assess HR proficiency, predict DDRi effectiveness, and the need for a combined treatment strategy (e.g., PARPi with other DDRi or TKIs) [30,162,208,273,274]. Unfortunately, due to the limited number of clinical trials involving ATMi/ATRi/CHK1i/DNA-PKi, biomarkers indicating resistance to these DDRi are largely unknown. A few examples include: PGBD5 and Cdc25A depletion have been associated with ATRi resistance, and overexpression of ATP-binding cassette G2 (ABCG2) increased CC-115 resistance [275,276,277,278].

## 6. Selection of New GB Radiopharmaceuticals Targeting the DDR

In order to select suitable candidate radiopharmaceuticals capable of targeting DDR kinases for GB imaging and therapy, several factors need to be considered, such as biochemical and pharmacological characteristics, radiolabeling, and radionuclide half-life options, and the ability to cross the BBB. The latter is affected by the molecular weight, lipophilicity, polar surface area, and hydrogen bond donors of the inhibitor [43]. In order to identify those DDRi that have the potential to become suitable GB TRT agents, in-house selection criteria were applied to all the above mentioned DDRi studied in GB (listed in Table 2). Thereby four DDRi are suggested that could potentially be converted into novel TRT radiopharmaceuticals: AZD1390, Nedisertib (M3814), SAR-020106, and MK8776 (Figure 6).

These DDRi contain a halogen in an aryl position that could be a designated location for radiohalogenation, for example, using iodine-125 (Auger emitter), iodine-131 (beta emitter), or astatine-211 (alpha-particle emitter); and/or qualify for insertion of a chelator substituent to harbor a therapeutic radiometal.

Nucleophilic halogen exchange (iodine for iodine) reactions are regularly used for the incorporation of radioiodine into organic molecules, with inorganic salts (ammonium sulfate) or copper (II) salts often being added to catalyze the iodine exchange. Notably, a naturally stable isotope of astatine does not exist, and, therefore, halogen exchange using astatine-211 would require the iodo- or bromo- derivatives [279]. However, this approach is unable to yield a pure, astatinated product since the unreacted iodo- or bromo-starting compounds cannot be removed. Therefore, astatination reactions generally occur through electrophilic substitution reactions in the presence of oxidants, with a new method developed using the substitution of a dihydroxyboryl group [279,280]. When radiohalogenating the abovementioned DDRi, the effect that the larger halogen will have on the modified molecule may also result in altered biological properties.

The consideration of using a chelator would be that the increase in size, molecular weight, and the possible change in overall charge of the inhibitor could affect pharmacological properties (lipophilicity, metabolism, biological half-life, target binding), and especially for GB targeting, the BBB crossing [43]. Attachment of chelators to biomolecules is generally carried out through a nucleophilic reaction between a bifunctional chelating agent and a primary amine. Insertion of a chelator into the structure of a DDRi would require the replacement of the substituent on an N- or O-atom with a functionalized chelating agent through a variety of available reactions.

### 6.1. ATMi AZD1390

AZD1390, developed by AstraZeneca, is a highly potent ATMi (10.000-fold more specific for ATM than for other PIKK members) that blocks ATM autophosphorylation at Ser-1981 and phosphorylation of KAP1 at Ser-824 [281]. AZD1390 has been converted to a ^11^C-radiolabeled drug that showed good BBB penetration (1% ID at T_max_[brain] = 21 min) in healthy volunteers. Results on the aspects of safety, tolerability, and pharmacokinetics of [^11^C]-AZD1390 in combination with RT are expected by 2024 [29,59]. The fast localization of AZD1390 to the brain limits the use of AZD1390 in TRT to those therapeutic radionuclides that match its biodistribution characteristics. AZD1390 has a piperidine moiety, an isopropyl moiety, and fluorine at the ortho-position of ring two. There are no crystal structures of ATM reported to date, and the ATM model developed by Degorce et al. was used in this review for SAR rationalization [282]. SAR studies have reported the need for the 4-amino and 3-carboxamide derivative within the structure, as well as the importance of the internal hydrogen bond that is formed between this moiety and the bioactive conformation of ATM [282]. It has a potential radiohalogenation site at the fluoride atom at the ortho-position of ring two. Direct radiohalogenation can potentially add a therapeutic radioiodine or radioastatine to that position. A chelator could possibly replace the piperidinyl moiety; however, it sits within the hydrophobic pocket and would most likely affect binding [75,283].

### 6.2. DNA-PKi Nedisertib (M3814)

Nedisertib (M3814, Peposertib, MSC2490484A), developed by BioVision Inc., Milpitas, CA, USA is an orally bioavailable, highly potent, and selective DNA-PKi. Nedisertib was well-tolerated as monotherapy in AST patients, and two clinical trials are currently evaluating nedisertib (peposertib) in combination with chemo/RT. The maximum systemic concentration of nedisertib occurred between 1–2 h after administration. The BBB penetration capabilities are still under investigation (CilinicalTrials.gov Identifier: NCT04555577). Based on the structural interactions between nedisertib and the active site of the DNA-PK, both the quinazoline and morpholino moieties bind into the hydrophobic pocket, while the pyridazine ring rotates to have π-π interactions with the quinazoline plane [284]. The chloro-fluorobenzene ring in the active site is directed towards the N-lobe, thus potentially allowing radiohalogenation at positions one and three of the ring. However, it is noted that the fluorine points towards the hydrophobic pocket, and thus, radioiodination or radioastatination at this position might not be feasible. The binding model further indicates that the methoxy group on the pyridazine ring is orientated outwards towards the solvent region. The methyl group could potentially be amended to a longer alkyl chain that will extend further into the solvent area and be functionalized with a chelator group in the terminal position. A chelator would be able to complex metallic radioisotopes for TRT.

### 6.3. CHK1i SAR-020106 and MK-8776 (SCH900776)

The kinase domain of CHK1/2 consists of an N- and C-terminal lobe with a hinge region connecting the two lobes. The hinge forms the ATP-binding pocket, and the majority of CHK1i will compete with ATP for binding to this site. Inhibitors bind through hydrogen bonding to peptides (typically Glu-85, Tyr-86, and Cys-87), as well as peptide-bound water within the active site. Generally, polar substituents of the inhibitors are orientated into the ribose pocket, with more lipophilic groups being directed toward the surface where the hinge cleft opens to the solvent. A substituent projecting into the solvent area could be modified with more hydrophilic groups in order to improve inhibitor pharmacokinetics [163].

SAR-020106 is a highly selective and potent inhibitor of CHK1 (IC_50_ of 13 nmol/L; >7 000-fold selectivity over CHK2) that is still in its preclinical phase. Although SAR-0020106 is highly bound (94%) to plasma proteins, the tumor drug accumulation within 24 h is significant, with tumor/plasma ratios of 47:1 and 85:1 after 6 h and 24 h, respectively [285]. SAR-020106 is structurally classified into the ‘pyrazine scaffold’ inhibitor group, with an ether-linked ethylamine substituent on a cyanopyrazine ring connected to a chlorinated isoquinoline [163]. The cyanopyrazine group significantly interacts with Lys-38 and the protein-bound water network within the active site, while the isoquinoline nitrogen and secondary amine connect with Cys-87 and Glu-85, respectively. The chloro-group on the isoquinoline indicates a potential position for radiohalogenation using radioiodine or –astatine since this chlorine atom is not involved in active site interaction. The nitrogen atom of the tertiary amine side chain of SAR-020106 also binds to water within the protein active site, but this amine has two methyl groups, one of which could potentially be substituted with a longer alkyl chain extending into the solvent region. A lengthened alkyl chain should not drastically affect the hydrogen bonding of the amine and would potentially allow for the insertion of a chelating group at the end of the hydrocarbon chain. The chelator could then be used for the complexation of therapeutic metal isotopes, such as lutetium-177, for TRT.

MK-8776 (SCH900776), developed by Merck KGaA, is another highly selective and potent inhibitor of CHK1 (IC_50_ of 3 nmol/L) that is currently in phase I/II clinical trials for various cancers but has only been tested preclinically for GB therapy [286,287]. These studies have indicated that MK-8776 enhances cellular susceptibility to chemotherapeutic agents, such as gemcitabine and hydroxyurea [288]. The BBB penetration of MK-8776 is currently unknown, but the drug is 49% plasma protein bound with a plasma half-life of 5.6–9.8 h [164]. The structural scaffold for MK-8776 is pyrazolo[1,5-a]pyrimidine functionalized with a piperidine and 1-methyl-pyrazole ring [163,289]. MK-8776 binds to the hinge region of the kinase ATP-binding site through N1 and C7-NH_2_ of the pyrazolo[1,5-a]pyrimidine core, while the nitrogen of the 1-methyl pyrazole is within the interior pocket bound to water. The piperidine nitrogen atom is hydrogen-bonded to Glu-91 and the amide carbonyl of Glu-134 in the ribose pocket. Position C6 of the pyrazolo[1,5-a]pyrimidine is functionalized with bromine which could potentially be converted to a therapeutic radiohalogen. Although the C7 primary amine of MK-8776 is involved in binding to the active site, similar compounds with a secondary amine in this position that were investigated prior to the development of the clinical candidate also indicated very selective and strong inhibition of CHK1 [163]. Therefore, alkylation of the C7-amine with a chelator-functionalized alkyl chain (to harbor therapeutic metal radionuclides) will convert MK-8776 into a TRT-radiopharmaceutical.

## 7. Conclusions

DDR kinases are attractive targets to promote DNA damage and DNA replication stress and to render GB cells more vulnerable to RT and TMZ, following the principle of synthetic lethality. The current DDRi targeting ATM/ATR, PARP, CHK1/2, and DNA-PK for the treatment of GB and a perspective and overview on potential radiolabeling options for those small molecules are presented. Despite the hurdles of GB heterogeneity and drug resistance, radiopharmaceuticals targeting DDR kinases have the potential to stratify patients for DDRi therapy, predict response to DNA damaging treatments and guide TRT agents to the nucleus of GB cells, ultimately increasing therapeutic effectiveness. This review revealed that only a limited number of developed DDRi have been explored for their TRT potential. Through the application of relevant selection criteria, four DDRi compounds were identified that could potentially be converted into novel TRT radiopharmaceuticals: AZD1390, Nedisertib (M3814), SAR-020106, and MK8776. Radiopharmaceutical development of these candidates may greatly influence a more tailored and personalized GB therapy.

## Figures and Tables

**Figure 1 cancers-14-01821-f001:**
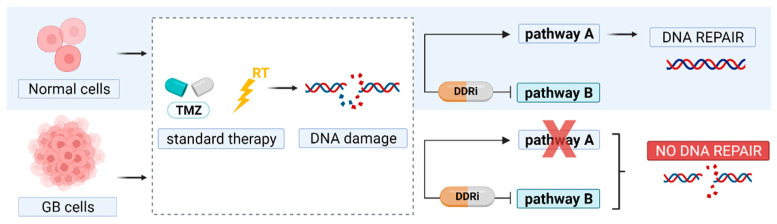
Principle of synthetic lethality. In glioblastoma (GB) therapy, DNA damage is induced by temozolomide (TMZ) and radiation therapy (RT). DNA repair pathways are often disrupted (pathway A) and therefore GB cells solely depend on a back-up pathway to repair DNA damage. Inhibitors of essential DNA damage response kinases (DDRi) can block this rescue pathway to promote GB cell death.

**Figure 2 cancers-14-01821-f002:**
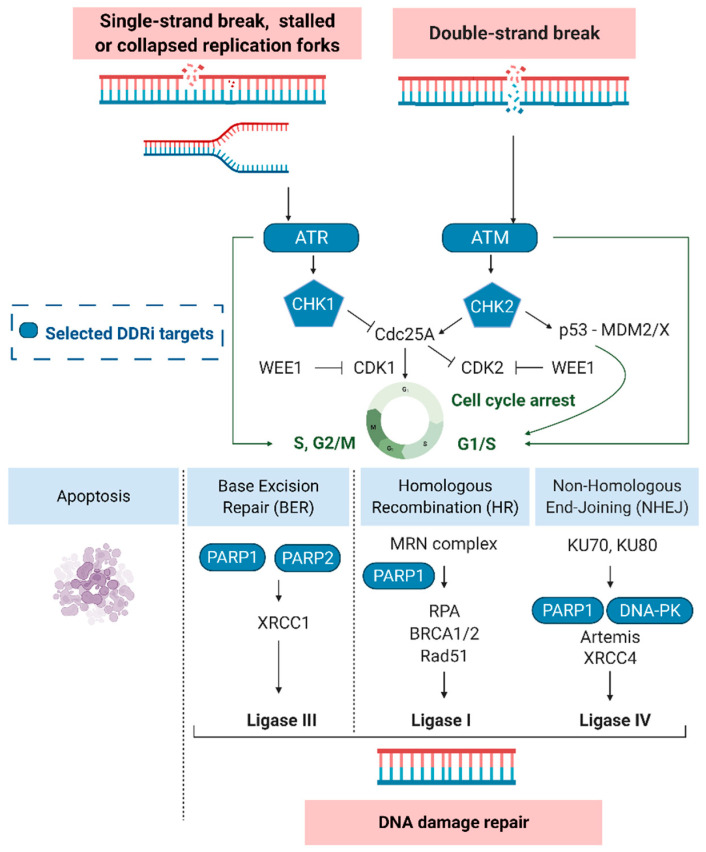
The DNA damage response and selected targets (blue). Ataxia-telangiectasia mutated (ATM), ATM-RAD3-related protein (ATR), cyclin-dependent kinase 1 (CDK1/2), checkpoint kinase-1 and -2 (CHK1/2), DNA-dependent protein kinase (DNA-PK), Mouse double minute 2/X homolog (MDM2/X), Nbs1/hMre11/hRad50 (MRN complex), poly (ADP-ribose) polymerase-1 and -2 (PARP), replication protein A (RPA), X-ray repair cross-complementing protein 4 (XRCC4).

**Figure 3 cancers-14-01821-f003:**
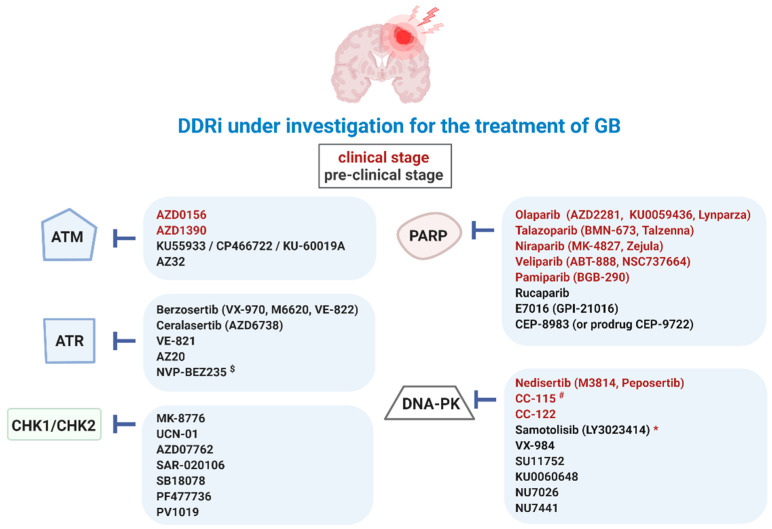
Overview of DDRi for targeted therapy of GB. Ataxia-telangiectasia mutated (ATM), ATM-RAD3-related protein (ATR), checkpoint kinase-1 and -2 (CHK1/2), DDR kinase inhibitor (DDRi), DNA-dependent protein kinase (DNA-PK), poly (ADP-ribose) polymerase-1 and -2 (PARP). ^$^ also targets phosphatidylinositol-3-kinase (PI3K)/Akt and the mammalian target of rapamycin (mTOR) signaling pathways (PI3K/mTOR), * paediatric central nervous system tumors, ^#^ also targets mTOR.

**Figure 4 cancers-14-01821-f004:**
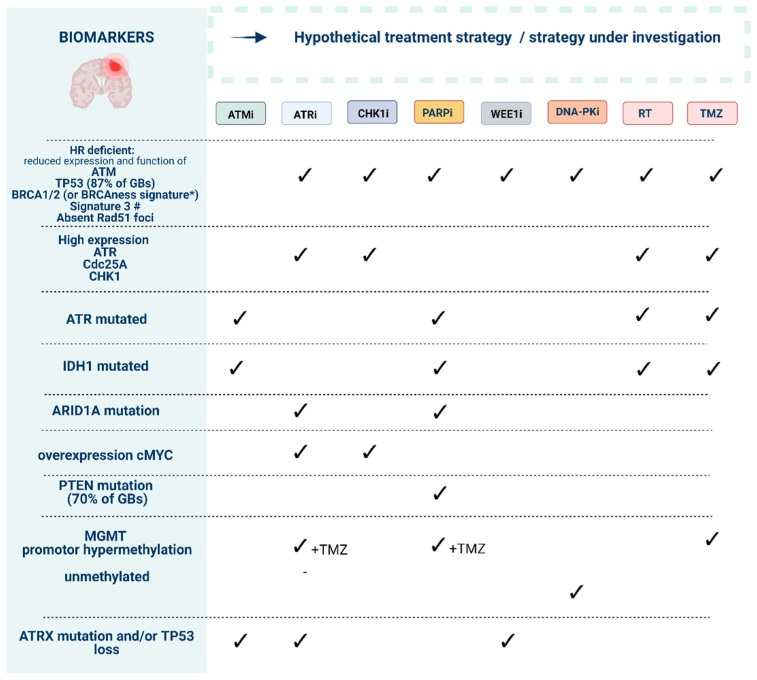
Preliminary biomarkers to guide the use of DNA damage response (DDR) inhibitors in order to reach synthetic lethality. Ataxia-telangiectasia mutated (ATM), ATM-RAD3-related protein (ATR), checkpoint kinase-1 and -2 (CHK1/2), DDR kinase inhibitor (DDRi), DNA-dependent protein kinase (DNA-PK), poly (ADP-ribose) polymerase-1 and -2 (PARP). * “BRCAness” signature can include mutations in *ATM*, *ATR*, *BAP1*, *BRCA1*, *BRCA2*, *CDK12*, *CHK1*, *CHK2*, *FANCA*, *FANCC*, *FANCD2*, *FANCE*, *FANCF*, *PALB2*, *NGS1*, *WRN*, *RAD50*, *RAD51B*, *RAD51C*, *RAD51D*, *MRE11A*, *BLM*, *BRIP1*. # Mutational signature found in human cancers characterized by defective homologous-recombination-based DNA double-strand break repair [33].

**Figure 5 cancers-14-01821-f005:**
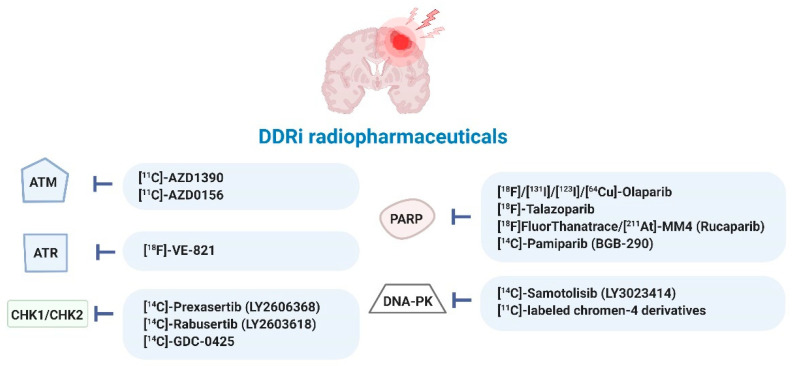
Radiopharmaceuticals targeting DDR kinases studied in various cancer types (for details see Appendix A). Ataxia-telangiectasia mutated (ATM), ATM-RAD3-related protein (ATR), checkpoint kinase-1 and -2 (CHK1/2), DDR kinase inhibitor (DDRi), DNA-dependent protein kinase (DNA-PK), poly (ADP-ribose) polymerase-1 and -2 (PARP) [29,40,49,50,52,53,54,55,56,57,58,59,60,61,62,63,64,65,66,67,68,69,70,71,72,73,74,75,76,77,78,79,80,81,82,83,84,85,86,87,88,89,90,91].

**Figure 6 cancers-14-01821-f006:**
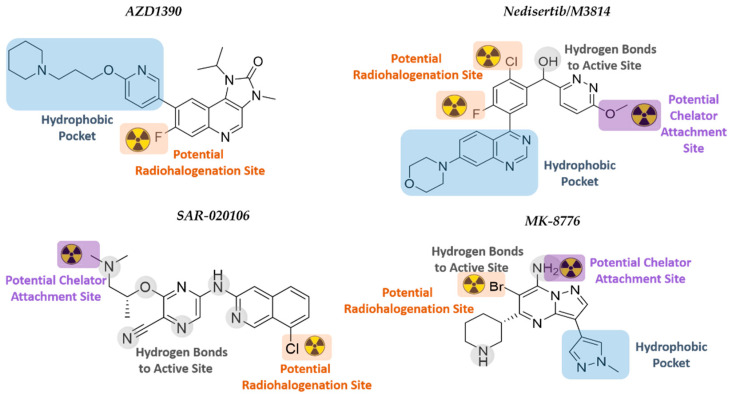
Chemical structures of the selected DDRi. The structure–activity relationship and potential radionuclide attachment sites are indicated.

**Table 1 cancers-14-01821-t001:** Relevant clinical trials concerning DDRi therapy in glioma patients.

Target	Drug	Combined Therapy *	Clinical Phase	Glioma Type	Biomarker Selection/Evaluation	Trial Number (Status) Reference
ATM	AZD1390	RT	I	nd/rec GB, malignant brain neoplasms	-	NCT03423628 (r)
ATM	AZD0156	Olaparib/Irinotecan/Fluorouracil/Folonic acid	I	AST (incl glioma)	-	NCT02588105 (anr)
PARP1/2/3	Olaparib	TMZ/RT	II	advanced	IDH1/2 mutations	NCT03212274 (r)
		TMZ/RT	II	rec HGG	IDH mutant	NCT03561870 (unknown)
		Cediranib/vs Bevacizumab	II	rec GB	Angiogenesis-DNA repair	NCT02974621 (anr)
		TMZ	I	rec GB	-	NCT01390571 (c) [129]
		Pamiparib/TMZ/RT	0/I	nd/rec GB	-	NCT04614909 (r)
PARP1/2	Veliparib (ABT-888)	TMZ	II/III	nd GB	methylated MGMT-DDR genes-MGMT-PARP1	NCT02152982 (anr)
		TMZ/RT	II	nd GB	unmethylated MGMT	[130]
		TMZ/RT	II	nd grade III-IV	no H3 K27M/BRAFV600 mutations	NCT03581292 (anr)
		TMZ	I/II	rec GB	-	[131]
		TMZ/RT	I	nd GB	plasma proteomic evaluation	NCT00770471 (c)
PARP1/2	Talazoparib	Carboplatin	II	rec GB	IDH/PTEN mutation “BRCAness” signature *	NCT04740190 (r)
PARP1/2	Niraparib (MK-4827)	RT	II	rec GB	-	NCT04715620 (r)
		Tumor-Treating Fields	II	rec GB	MGMT	NCT04221503 (r)
		TMZ	I	advanced cancer, incl GB	-	NCT01294735 (c) [132]
PARP1/2	Pamiparib (BGB-290)	TMZ/RT	I/II	nd/rec GB	MGMT	NCT03150862 (c)
		TMZ	I/II	rec grade II-IV	IDH1/2-mutant	NCT03914742 (r)
		TMZ	I	nd/rec grade I-IV	IDH1/2-mutant	NCT03749187 (r)
		TMZ/RT	0/I	nd/rec GB	-	NCT04614909 (r)
DNA-PK	Nedisertib (M3814)	TMZ/RT	I	nd GB	MGMT unmethylated	NCT04555577 (r)
PI3K/mTOR/DNA-PK	Samotolisib (LY3023414)	-	II	paediatric CNS tumors	PI3K/mTOR mutations	NCT03213678 (r)

Abbreviations: active non-recruiting (anr), advanced solid tumors (AST), central nervous system (CNS), (c) completed, high-grade glioma (HGG), isocitrate dehydrogenase(IDH), newly diagnosed (nd), O^6^-methylguanine-DNA methyltransferase (MGMT), poly(ADP-ribose)polymerase (PARP), phosphatidylinositol-3-kinase and the mammalian target of rapamycin (PI3K/mTOR), phosphatase and tensin homolog (PTEN), recruiting (r), recurrent (rec), * “BRCAness” signature (*ATM*, *ATR*, *BAP1*, *BRCA1*, *BRCA2*, *CDK12*, *CHK1*, *CHK2*, *FANCA*, *FANCC*, *FANCD2*, *FANCE*, *FANCF*, *PALB2*, *NGS1*, *WRN*, *RAD50*, *RAD51B*, *RAD51C*, *RAD51D*, *MRE11A*, *BLM*, *BRIP1*).

**Table 2 cancers-14-01821-t002:** Selection criteria for assessment of candidate GB DDRi TRT agents.

**Inclusion Criteria**
1. The DDRi was studied preclinically or in clinical trials in GB. 2. The DDRi is a small molecule that: A. contains a halogen which indicates a position that can potentially be radio-iodinated or -astatinated; and/or B. has a potential site for attachment of a chelator. 3. The DDRi has already been radiolabeled with a diagnostic isotope and was studied in GB.
**Exclusion Criteria**
1. Clinical trials results indicate candidate exclusion by way of: A. findings in GB patients revealed unwanted safety/tolerability issues (single agent), serious adverse events that were irreversible or responsible for treatment discontinuation, and/or B. occurrence of unfavorable pharmacokinetic properties. 2. The DDRi does not contain a halogen or any possible site for chelator attachment. 3. The DDRi has already been radiolabeled (diagnostic and/or therapeutic radionuclide) but was not studied in GB.

## Data Availability

The data presented in this study are available in the article and Appendix A.

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
