# Peer review of "Perspective on the Use of DNA Repair Inhibitors as a Tool for Imaging and Radionuclide Therapy of Glioblastoma"

_cancers, 2022, doi:10.3390/cancers14071821_

Round 1

Reviewer 1 Report

  • lapsus, I think; in text is referred to Table S1, Table S2 - but you cannot find this back in the manuscript, you can find Table 1 and Table 2,
  • p. 7 to 14/34, authors referring to the different DDR pharmaceuticals- and radiopharmaceuticals, then - for my feelings - paraphs 4, 5, 6, and 7, should be ''sub-points'', to paraph [3] !
  • lines 301-306, under Table 1, to what refers this ? 
  • line 608; not better to state : ''challenging and risks of DDRi pharmaceuticals - and radiopharmaceuticals''

Author Response

Response to Reviewer 1 Comments

Point 1: lapsus, I think; in text is referred to Table S1, Table S2 - but you cannot find this back in the manuscript, you can find Table 1 and Table 2

Table 1 and Table 2 are indeed included in the manuscript while Table S1 and Table S2 can be found in supplementary materials.

Table S1: Overview of cancer clinical trials of ATMi and ATRi

Table S2: Cancer radiopharmaceuticals targeting DDR kinases

Point 2: p. 7 to 14/34, authors referring to the different DDR pharmaceuticals- and radiopharmaceuticals, then - for my feelings - paragraphs 4, 5, 6, and 7, should be ''sub-points'', to paraph [3] !

In the original structure, paragraph 3 includes a perspective on the potential of radiopharmaceuticals targeting DDR kinases and an overview of the ones studied in various cancer types. Original paragraphs 4-5-6-7 include 2/3 main points: rationale of the target selection for GB therapy, current status of the ‘cold’ DDRi and current radiopharmaceuticals. We agree with the reviewer to adapt the structure and adapted the title of section 3 to DDR (radio)pharmaceuticals:

new structure:

  1. DDR (radio)pharmaceuticals

3.1. ATM/ATR inhibitors

3.1.1. Targeting ATM/ATR as an anti-GB strategy

3.1.2. Current status of ATM/ATR targeted therapy in GB

3.1.3. ATM/ATR radiopharmaceuticals

3.2. CHK1/2 inhibitors

3.2.1. Current status of CHK1/2 targeted therapy in GB

3.2.2. CHK1/2 radiopharmaceuticals

3.3. PARP inhibitors

3.3.1. Current status of PARP targeted therapy in GB

3.3.2. PARP radiopharmaceuticals

3.4. DNA-PK inhibitors

3.4.1. Current status of DNA-PK targeted therapy in GB

3.4.2. DNA-PK radiopharmaceuticals

Point 3: lines 301-306, under Table 1, to what refers this ?

Notes were added under the table to add all abbreviations in full that were used in the table. Biomarkers included in the * "BRCAness" signatures were also listed.

Point 4: line 608; not better to state : ''challenging and risks of DDRi pharmaceuticals - and radiopharmaceuticals''

The title of section 9 (Line 621) was changed from “Challenges and risks of DDRi and DDRi radiopharmaceuticals” to “Challenges and risk of DDRi (radio)pharmaceuticals”.

Reviewer 2 Report

The authors review the use of radiopharmaceuticals targeting DNA damage repair kinases (DDRi) for molecular imaging and targeted radionuclide therapy (TRT) in glioblastoma (GB).
The manuscript is well written and organized with a clear introduction that also outlines the topics that are covered.
Despite being a long document, the authors address in concise fashion the main topic. Firstly, they provide insights on the targeting pathway of DDR in GB with particular emphasis on four targets and highlight other hypothetical treatment strategies.
Secondly, the overview of radiotracers developed for diagnostic and TRT includes several inhibitors (ATM/ATR, CHK1/2, PARP).  
Finally, the authors provide some insights on the challenges to develop new DDR radiotracers towards four targets mentioned above and what are the important parameters to design new radiotracers for diagnostic and TRT in GB.

Overall, the manuscript is very important to cancer readers and the radiopharmaceutical community. 

Author Response

We thank the reviewer for his/her positive comments and his/her acknowledgement of the importance of this work to the oncology and radiopharmaceutical community.

Reviewer 3 Report

In this review the authors review various therapies used for glioblastomas, mainly the ones that target the DDR. The review is comprehensive and well written. I have only a few minor comments concerning rewriting or re-explaining some sections.

Figure 2 is somewhat misleading. For example, ATR signals replication damage (e.g. stalled or collapsed replication forks) and as it is diagramed now it is not clear what it does. In fact, at line 239 it is listed as signaling single strand breaks which is an oversimplification. It is actually the major S-phase checkpoint kinase. Additionally, MRN (PubMed ID33339169) and KU70/80 are also considered break sensors (PubMed30177438). The diagram is OK, but they are discussed in the text as sensors (line 103). In fact, Ku70/80 heterodimer has very high affinity for broken ends and is believed to bind to broken ends first. There is then competition between Ku70/80 and MRN depending on whether repair will proceed by HR or NHEJ. This depends on the cell cycle. The authors should also expand their discussion to include DNA replication stress which is believed to be targeted in fast dividing cancer cells.

Line 122: There is actually correlation between IDH1 mutation and DNA damage checkpoint kinases (PMID: 34503108). The authors should explore this in some more detail. In fact, gliomas are classified as IDH WT or IDH mutant. The two cancers are treated differently because IDH mutant has a better prognosis. The authors should rewrite some of the text to reflect this.

Author Response

Response to Reviewer 3 Comments

Point 1: Figure 2 is somewhat misleading. For example, ATR signals replication damage (e.g. stalled or collapsed replication forks) and as it is diagrammed now it is not clear what it does. In fact, at line 239 it is listed as signaling single strand breaks which is an oversimplification. It is actually the major S-phase checkpoint kinase. Additionally, MRN (PubMed ID33339169) and KU70/80 are also considered break sensors (PubMed30177438). The diagram is OK, but they are discussed in the text as sensors (line 103). In fact, Ku70/80 heterodimer has a very high affinity for broken ends and is believed to bind to broken ends first. There is then competition between Ku70/80 and MRN depending on whether repair will proceed by HR or NHEJ. This depends on the cell cycle.

ATR is indeed also activated by single stranded DNA structures, which may for example arise at resected DNA DSBs or stalled replication forks. Hence, we agree with the reviewer and added this information to the manuscript (Line 247). This was also implemented in Figure 2.

ATR is indeed a main S-phase checkpoint kinase and also a principal mediator of the G2/M cell cycle checkpoint. In contrast, ATM plays a crucial role in the activation of the G1/S cell cycle checkpoint, which prevents cells with damaged DNA from entering S-phase (primarily mediated through p53). We agree to add this info to the manuscript. The following was added (Line 252): “However, ATM plays a crucial role in the activation of the G1/S cell cycle checkpoint while ATR enforces the intra-S-phase and G2/M cell cycle checkpoint”. Extra arrows were implemented in Figure 2 to highlight the cell cycle checkpoint functions (see green arrows). Another arrow was added to the figure from p53 to the cell cycle logo (pubmed 25512053).

We also agree with the reviewer that the MRN complex and KU70/80 are both break sensors. The publication of Shibata et al. indeed showed that Ku has exquisite end-binding capacity, is the first protein binding most DSBs and plays a role in determining pathway choice. Extra info on this was added to the manuscript and the reference of Shibata et al. was added: “The DNA-PK and MRN complexes assemble and compete at sites of DNA DSBs where they act as damage sensors and initiate cell cycle dependent NHEJ or HR, respectively.” (Line 110).

Figure 2 was adapted (see also attached PDF).

Point 2: The authors should also expand their discussion to include DNA replication stress which is believed to be targeted in fast dividing cancer cells.

We agree with the reviewer that the presence of DNA replication stress in cancer cells was currently not included and needs to be mentioned. The following was added to the introduction (Line 77): “Interestingly, ‘replication stress’ present in cancer cells could further be enhanced following these therapies through further loosening the remaining checkpoints and inducing failure of further proliferation”.

Also added (Line 633): The phenomenon of “replication stress”, unique to fast proliferating cancer cells, enforces this statement.

In the conclusion (Line 821), DNA replication stress was added.

This reference was added (now nr 16): https://www.ncbi.nlm.nih.gov/pmc/articles/PMC4999839/.

Point 3: Line 122: There is actually correlation between IDH1 mutation and DNA damage checkpoint kinases (PMID: 34503108). The authors should explore this in some more detail. In fact, gliomas are classified as IDH WT or IDH mutant. The two cancers are treated differently because IDH mutant has a better prognosis. The authors should rewrite some of the text to reflect this.

Thank you for this interesting comment. These correlations are indeed important since the IDH mutation status plays a big role in the current therapy decision process in GB patients. We added the following to the manuscript (Line 129-133): “Interestingly, co-mutations in DDR kinases could play a role. In IDH1 mutated astrocytoma patients, TP53 (63%) and ATRX (27%) are the top two genes that display a higher frequency of mutations. An association between IDH1 mutations and reduced ATRX expression has also been shown. Mutations in CHK2 are instead associated with an IDH1-wildtype astrocytoma.”
